# Syntheses, Structures and Reactivity of Metal Complexes of Trindane, Trindene, Truxene, Decacyclene and Related Ring Systems: Manifestations of Three-Fold Symmetry

**DOI:** 10.3390/molecules28237796

**Published:** 2023-11-27

**Authors:** Philippa E. Lock, Nada Reginato, Julia Bruno-Colmenárez, Michael J. McGlinchey

**Affiliations:** School of Chemistry, University College Dublin, Belfield, D04 V1W8 Dublin, Irelandjulia.bruno@ucd.ie (J.B.-C.)

**Keywords:** ketone triple condensations, polycyclic frameworks, metal π-complexes, C_60_ fragments, three-fold symmetry, X-ray crystallography

## Abstract

The triple condensation of cyclopentanone or indanone to trindane (C_15_H_18_) or truxene (C_27_H_18_), respectively, provides convenient access to molecular skeletons on which major fragments of the prototypical fullerene C_60_ can be assembled. In particular, early approaches (both organic and organometallic) towards sumanene, as well as the final successful synthesis, are described. Organometallic derivatives of trindane have been prepared in which Cr(CO)_3_, Mo(CO)_3_, [Mn(CO)_3_]^+^ or [(C_5_H_5_)Fe(CO)_2_]^+^ are η^6^-bonded to the central arene ring. The debromination of hexabromotrindane yields trindene, which forms a tri-anion to which as many as three organometallic fragments, such as Mn(CO)_3_, W(CO)_3_Me, or Rh(CO)_2_, may be attached. Truxene forms complexes whereby three metal fragments can bind either to the peripheral arene rings, or to the five-membered rings, and these can be interconverted via η^6^ ↔ η^5^ haptotropic shifts. Truxene also forms a double-decker sandwich with Ag(I) bridges, and decacyclene, C_36_H_18_, forms triple-decker sandwiches bearing multiple cyclopentadienyl-nickel or -iron moieties. The organic chemistry of trindane has been investigated, especially with respect to its unexpectedly complex oxidation products, which were only identified unambiguously via X-ray crystallography. The three-fold symmetric trindane framework has also been used as a template upon which a potential artificial receptor has been constructed. Finally, the use of truxene and truxenone derivatives in a wide range of applications is highlighted.

## 1. Introduction

The cyclopentadienide fragment is a component of many polycyclic systems, and their metal complexes are very numerous [1]. We focus here on the syntheses, structures and reactivity of three-fold symmetric ligands, such as trindane or trindene, and related frameworks whereby the molecular periphery has been augmented via the incorporation of additional benzo rings, as in truxene or decacyclene.

Trindane, tris(cyclopenteno)benzene, **1**, first reported by Wallach in 1897 [2], is commonly prepared via the acid-catalysed condensation of three molecules of cyclopentanone [3], and was characterised via X-ray crystallography in 1964 [4]. Other trimerisation routes include the use of SiCl_4_ in ethanol at room temperature [5], TiCl_4_ in refluxing ethanol [6], or aqueous NH_4_Cl at high temperatures [7]. The bromination of trindane via the photolysis of Br_2_ in CCl_4_ gave hexabromotrindane, **2**, [8,9], which upon debromination with zinc furnished both isomers of dihydro-1*H*-trindene, **3**. Subsequent multiple deprotonation (Figure 1) generated the trindene trianion, **4**, ideally poised for reactions with a range of organometallic fragments.

## 2. Organometallic Derivatives of Trindane

### 2.1. Trindane Complexes of Chromium, Molybdenum, Manganese and Iron

The earliest report of an organometallic derivative of **1** appears to describe the formation of a series of cationic complexes of the type [^99m^Tc(arene)_2_]^+,^, where the arene was C_6_H_6_, C_6_Me_6_, C_6_Et_6_, indane, trindane, etc. [10]. These materials were injected into rats to study the biodistribution of radioactive technetium to test their viability as myocardial imaging agents. These studies were, out of necessity, carried out on trace quantities of material, and no analytical or spectroscopic data were reported for [^99m^Tc(trindane)_2_]^+^ [PF_6_]^−^, **5**. In terms of the first fully characterised derivatives, complexes of the type (η^6^-trindane)ML_n_, where ML_n_ = Cr(CO)_3_, **6**, Mo(CO)_3_, **7**, [Mn(CO)_3_]^+^, **8**, or Fe(C_5_H_5_)^+^, **9**, were each prepared via the reaction of trindane with an appropriate organometallic precursor (Figure 2) [11].

(η^6^-Trindane)Cr(CO)_3_, **6**, forms yellow crystals, the structure of which appears in Figure 1, and reveals that the tripodal moiety is oriented such that the three carbonyl ligands are staggered with respect to the cyclopentenyl rings. These five-membered rings adopt envelope conformations such that the three methylene “wingtips” are folded in an *endo* fashion relative to the metal [11]. This may be compared to the structures of the previously known molecules [tris(cyclohexene)benzene]ML_n_, where ML_n_ = Cr(CO)_3_ or [Mn(CO)_3_]^+^, **10**, which also crystallise in a staggered tripodal orientation, but their peripheral six-membered rings exhibit conventional half-chair conformations [12].

It was noteworthy that in the mass spectra of the neutral complexes of (η^6^-trindane)M(CO)_3_, where M = Cr or Mo, peaks with *m*/*z* values corresponding to those of [(trindane)_2_M_2_(CO)_3_]^+^ were observed, leading to the speculation of the formation of triple-bridged systems of the type [(η^6^-trindane)M(μ-CO)_3_M(η^6^-trindane)]^+^, **11**, entirely analogously to known 30-electron systems such as (η^5^-C_5_Me_5_)Re(μ-CO)_3_Re(η^5^-C_5_Me_5_) [13].



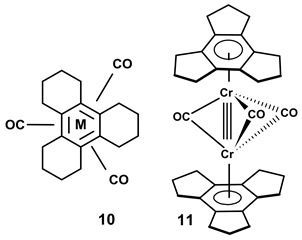



Trindane itself exhibits a very simple ^1^H NMR spectrum, i.e., a triplet (12H) for the benzylic protons and a quintet (6H) for the wingtip methylene groups. The incorporation of a π-complexed organometallic fragment renders the faces of the trindane ligand inequivalent, thus giving rise to four ^1^H NMR environments in complexes such as **6** through **9**. The benzylic protons (each 6H) are readily distinguished from the resonances for the wingtip methylenes (each 3H) by their relative intensities, but their assignment to *exo* or *endo* positions is less trivial. However, the X-ray data for the chromium complex, **6**, provide a rational means of assigning these resonances. Figure 2 illustrates the dihedral angles between a pair of benzylic hydrogens and those of the neighbouring wingtip methylene group within a five-membered ring. In all cases, the *endo*-benzylic hydrogen and its adjacent *exo*-wingtip hydrogen make a dihedral angle of approximately 90°, which is typical of di-equatorial interactions, which, in accordance with the Karplus relationship relating dihedral angles to ^3^*J*_H-H_ values, exhibit a rather small vicinal coupling constant (4–5 Hz). In contrast, the diaxial interaction between the *exo*-benzylic hydrogen and its *endo*-wingtip counterpart leads to a noticeably larger ^3^*J*_H-H_ value (~12 Hz), thus allowing a straightforward assignment of all the resonances.

### 2.2. Trindane Complexes of Ruthenium

An arene exchange reaction between [(p-cymene)RuCl_2_]_2_ and molten trindane was attempted and, gratifyingly, [(trindane)RuCl_2_]_2_ was produced in a 85% yield; its structure appears in Figure 3. In the solid state, the complex adopts the arrangement whereby the two ruthenium atoms are linked by two bridging chlorines, as in **12a**, and each also possesses a terminally bonded chlorine. However, in solution, ^1^H-^1^H and ^1^H-^13^C two-dimensional NMR data clearly reveal the existence of a second species, the triple-bridged ionic isomer [(η^6^-trindane)Ru(μ-Cl)_3_(η^6^-trindane)]Cl, **12b**, formed via the loss of a chloride ligand. These isomers are in a temperature-dependent equilibrium such that the ratio of **12a** to **12b** is 30:70 at −50 °C in CD_2_Cl_2_, but at room temperature in nitromethane CD_3_NO_2_ (which would surely favour the ionised species), this ratio changes dramatically to 1:10 [14].

We note parenthetically that hexaethylbenzene (HEB) may be thought of as an “unrestricted” trindane in which the wingtip methylene groups of the cyclopentenyl rings are now untethered, thus leaving them free to rotate. It was shown that the analogous ruthenium complex of HEB, i.e., [(HEB)RuCl_2_]_2_, also exists as an interconverting double- and triple-bridged species, but in this case both the double-bridged neutral species, **13a**, analogous to **12a**, and also the *D*_3h_-symmetric cation, **13b**, analogous to **12b**, together with its [C_5_(CO_2_Me)_5_]^−^ counter-anion, have been characterised via X-ray crystallography (Figure 4) [15,16,17].

When [(trindane)RuCl_2_]_2_, **12**, was allowed to react with excess trindane in the presence of AgBF_4_, the sandwich compound [(η^6^-trindane)_2_Ru]^2+^ 2[BF_4_]^−^, **14**, was isolated and fully characterised spectroscopically [14]. This molecule is, of course, the ruthenium analogue of the [^99m^Tc(trindane)_2_]^+^ [PF_6_]^−^ complex, **5**, claimed, but never unambiguously identified, in the medicinal study of potential radio-labelled myocardial imaging agents noted above [10]. Finally, it was found that treatment of **12** with trimethyl phosphite delivered the anticipated monomeric species (η^6^-trindane)RuCl_2_[P(OMe)_3_], **15**, the structure of which is shown in Figure 5.

## 3. Organometallic Derivatives of Trindene

As noted above, Katz and Ślusarek prepared the trindene trianion, **4**, which was then allowed to react with ferrous chloride to form bis(trindene)diiron, **16**, as a red-brown material, along with traces of a tri-iron complex [8]. NMR data suggested that that the hydrocarbon rings in **16** are disposed as *anti*, as depicted in Figure 3.

Almost four decades later, the first triple-metallocene derivative of trindene was prepared via the exchange reaction of the potassium salt of **4** with [Fe(C_5_H_5_)(η^6^-fluorene)]PF_6_ to yield the *syn*,*syn*,*syn*- and *syn*,*syn*,*anti*-isomers of (η^5^:η^5^:η^5^-trindenyl)[Fe(C_5_H_5_)]_3_, **17a** and **17b**, respectively, in a 1:3 ratio. The latter molecule was characterised via X-ray crystallography (Figure 6), which clearly showed a notable deviation of two five-membered rings in the trindene skeleton from planarity caused by the need to relieve repulsive non-bonding interactions between the *syn* ferrocenyl groups [18].

The organometallic chemistry of trindene was considerably extended by Lynch and Rheingold, who successfully characterised a series of trimetallic derivatives bearing manganese- or rhenium-tricarbonyl fragments. The treatment of dihydro-1*H*-trindene, **3**, with KH and either Mn(CO)_3_(py)_2_Br (py = pyridine) or [Re(CO)_3_(THF)Br]_2_ delivered (η^5^:η^5^:η^5^-trindenyl)[M(CO)_3_]_3_, where M = Mn or Re, **18** or **19**, respectively (Figure 4). As was found in **17b**, the X-ray crystal structure of **19** (Figure 7) revealed that the metals were situated in a *syn*,*syn*,*anti*-fashion such that the five-membered rings bearing the *cis* rhenium units were bent away from each other and out of the central plane by ~10°, and that the Re(CO)_3_ groups exhibited maximal staggering of their carbonyl ligands [19]. It was also established that when only two metal carbonyl moieties were introduced, they adopted a *trans* geometry, **20**, thus facilitating the introduction of a third *different* substituent, thereby forcing the *cis* disposition of the heterometals, as in the (trindenyl)Re_2_Rh(CO)_8_ complex, **21**.

In these systems, two of the metal fragments are necessarily positioned proximate to each other on the same face*,* and this feature has been exploited in a series of tri-molybdenum or tri-tungsten complexes. When the trindene trianion, **4**, was treated with Mo(CO)_6_ and then with methyl iodide, the major product was *syn*,*syn*,*anti*-(η^5^:η^5^:η^5^-trindenyl)[Mo(CO)_3_Me]_3_, **22**, along with a lower quantity of a material, **23**, in which the two adjacent molybdenum atoms had lost their methyl substituents and formed a metal–metal bond. The tungsten congener behaved similarly, and the structure of the corresponding benzyl analogue, **24**, possessing a tungsten–tungsten linkage was verified via X-ray crystallography, as shown in Figure 8 [20].

The electrochemical behaviour of a number of these trimetallic derivatives of trindene was studied via cyclic voltammetry (CV). It was found that the complexes (trindenyl)(ML_n_)_3_, where ML_n_ = Mn(CO)_3_, **18**, or Rh(1,5-cyclooctadiene), **25**, showed three well-defined, reversible one-electron oxidation reactions. In the rhodium case, the observed formal potentials are −0.33, −0.16 and +0.43 V, for the couples **25**–**25**^+^, **25**^+^–**25**^2+^ and **25**^2+^–**25**^3+^, respectively, and this abnormally large separation in potentials has been interpreted in terms of a Class III (totally delocalised) system [21]. There is also a report of the tri-rhodium complex, (η^5^:η^5^:η^5^-trindene)[Rh(1,5-COD)]_2_Rh(CO)_2_, **26**, whose structure is shown in Figure 9 [22]. Since the two Rh(COD) fragments occupy *anti* positions, one can assume that they were coordinated initially, and the Rh(CO)_2_ unit was added subsequently.

The CV scan of the triferrocenyl system **17b** proceeds in two well-defined and reversible (chemically and electrochemically) one-electron steps with values of 0.23 and 0.61 V, for the first and second oxidations; however, this is followed by a chemically irreversible process (0.94 V) showing that the trication precipitates on the Au electrode [18]. Finally, we note that the charge transfer properties of multi-ferrocenyl dihydro-1*H*-trindenes, **27**, have been investigated [23].



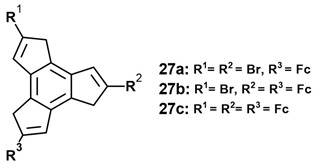



## 4. Metal Complexes of Truxene

Using the analogy of the self-condensation of cyclopentanone to form trindane, the analogous reaction of indanone yields *C*_3h_-symmetric truxene, C_27_H_18_, **28**. It was first obtained accidentally by Hausmann in 1889 [24] during the synthesis of indanone from 3-phenylpropanoic acid, but the mechanism involving the intermediate tetracyclic ketone shown in Figure 5 was only elucidated many years later [25].

The deprotonation of truxene with KH in the presence of Mn(CO)_3_(py)_2_Br yielded truxene–manganese adducts with one, two or three Mn(CO)_3_ units bound to five-membered rings of the truxenyl ligand, whereby in (η^5^:η^5^:η^5^-truxenyl)[Mn(CO)_3_]_3_, only the *syn*,*syn*,*anti*-isomer, **29a**, was obtained. However, these products can also be accessed via η^6^-to-η^5^ haptotropic shifts of arene-coordinated [Mn(CO)_3_]^+^ fragments in [(η^6^:η^6^:η^6^-truxenyl){Mn(CO)_3_}_3_]^3+^, which is formed as both the *syn*,*syn*,*anti*- and *all-syn* isomers, **30a** and **30b**, respectively; this provides the only route to the *all-syn* isomer **29b** (Figure 6) [26].

Truxene also reacts with silver perchlorate to form the dimeric sandwich compound [(μ^2^:μ^2^-truxene)_2_(η^1^-toluene)(H_2_O)Ag_2_(ClO_4_)_2_], **31**, in which the truxenes are linked by Ag(I) bridges, each bonded in a dihapto fashion to peripheral arene rings, as depicted in Figure 10 [27].

## 5. Metal Complexes of Decacyclene

Decacyclene, C_36_H_18_, **32**, was first obtained by Dziewonski in Krakow, Poland, in 1903 upon the dehydrogenation of acenaphthene with elemental sulphur at 205–295 °C [28]. It differs from truxene in terms of the incorporation of additional peripheral benzo rings, thus apparently recovering the *D*_3h_ symmetry of its trindenyl central core. However, the X-ray structure of decacyclene revealed it to be a shallow molecular propeller of *D*_3_ symmetry as the result of non-bonded repulsions between hydrogens of the peripheral naphthalene groups. This feature is clearly illustrated in the side view of **32** depicted in Figure 11. Interestingly, crystals of decacyclene also have a helical morphology, which is most pronounced when grown from solutions in organic solvents [29].

A more elegant route (Figure 7) has been reported that takes advantage of the palladium-catalysed trimerisation of strained cycloalkynes. The caesium fluoride-initiated elimination of fluorotrimethylsilane and triflate from the disubstituted acenaphthene, **33**, brought about the in situ formation of acenaphthyne, **34**, which underwent cyclisation to form decacyclene in a 23% yield [30].

Several triple-decker metal complexes of decacyclene have been reported from the Schneider group in Essen, Germany. The reduction of decacyclene in THF with potassium metal led to a red-brown solution containing polyanionic species that were allowed to react with organometallic reagents. Treatment with (η^5^-C_5_Me_4_Et)Ni(η^2^-acac) furnished black crystals of [(η^5^-C_5_Me_4_Et)Ni]_2_(μ-η^3^:η^3^)decacyclene], **35**, in which the two organo-nickel fragments were each bonded in an η^3^-fashion to opposite faces of the central arene ring (Figure 12) [31].

The analogous reaction with (η^5^-C_5_Me_4_Et)FeCl(tmeda), where tmeda is N,N,N′,N′-tetramethylethylenediamine, yielded two black crystalline products with the structures [(η^5^-C_5_Me_4_Et)Fe]_2_(μ_2_-η^6^:η^6^)decacyclene], **36**, and [(η^5^-C_5_Me_4_Et)Fe]_4_(μ_2_-η^6^:η^6^:η^6:^η^6^)decacyclene], **37**. In the former case, the organo-iron fragments were attached to opposite faces of the same naphthalene unit, as shown in Figure 8. Metal complexation in **37** occurs in an alternating up/down fashion on two naphthalene moieties, and the molecule adopts a gently twisting propeller geometry in accordance with the topology of the decacyclene skeleton (Figure 13). It is notable that the Fe-C distances to the bridging carbon atoms are longer than those to the others since the two electrons connecting them must be shared between the two metals, thereby satisfying the 18-electron rule [32].

## 6. Towards Sumanene

### 6.1. The Pyrolytic Approach

Although C_60_, with its “soccer-ball” icosahedral (*I*_h_) symmetry, is preparable via the evaporation of graphite under appropriate conditions, and is now commercially available, attempts to develop logical stepwise synthesis continue to be explored. Two major components of the C_60_ skeleton are corannulene C_20_H_10_, **38**, which possesses a central pentagon surrounded by five six-membered rings, and sumanene C_21_H_12_, **39**, which has alternating five- and six-membered rings around a central hexagon; both are illustrated in Figure 14. Corannulene is now available in kilogram quantities, thanks to the pioneering contributions of Larry Scott (in Nevada) and Jay Siegel (in Zurich), and this work has been comprehensively reviewed [33,34,35,36]. However, in 1997, when the organometallic chemistry of trindane was still in its early stages, no route to sumanene had been reported, and success was only achieved some years later (see below).

The bowl-shaped *C*_3v_-symmetric molecule sumanene, with its central six-membered ring surrounded by alternating five- and six-membered rings, was first postulated by Mehta in 1993; its name is derived from the Sanskrit word *suman*, which translates into “flower”, whereby the ring edges are considered to resemble petals. The earliest synthetic approach (Figure 9) involved the palladium-mediated thermolysis of *C*_3h_-symmetric 1,5,9-trimethyltriphenylene, **40**, in an attempt to bring about multiple dehydrogenations and subsequent ring closures; however, only the mono-bridged product, **41**, was formed. Under even more forceful conditions, the flash vacuum pyrolysis (FVP) of 1,5,9-tri(bromomethyl)triphenylene, **42**, led to the double-bridged species **43**, which was characterised via X-ray crystallography [37].

These experimental observations validate the computational results of Priyakumar and Sastry, which revealed that all these cyclopentenation steps lead to an increase in strain, with the third endothermic step leading to an increase of more than 50 kcal/mol, making the final ring closure almost impossible. In contrast, starting from trindane (thereby incorporating the five-membered rings early on in the procedure) and then generating the six-membered peripheral rings subsequently is a thermodynamically favoured process [38,39].

### 6.2. The Proposed Organometallic Approach

Retrosynthetic analysis (Figure 10) suggested a strategy starting from the cationic species [(trindane)Mn(CO)_3_]^+^, **8**, whereby multiple deprotonation at the benzylic positions with subsequent attack by an electrophile could lead to a system, **44**, bearing six bromomethyl substitutuents. The attachment of the organometallic fragment to the central arene ring would be expected to block that face and thereby favour *exo* attack by incoming electrophiles. Ring closure to form the bromosulfide **45**, oxidation to the corresponding sulfone, followed by a Ramberg–Bäcklund rearrangement (a standard technique for the formation of small rings [40]) of the triple halosulfone, **46**, should result in the elimination of HBr and SO_2_ and the direct formation of the new alkene moieties, leaving only a final dehydrogenation step.

This approach was predicated on earlier work by Astruc [41] on the multiple successive deprotonations at benzyl positions in [(η^6^-C_6_Me_6_)Fe(C_5_H_5_)]^+^, **47**, to initially form (η^5^-Me_5_C_6_=CH_2_)Fe(C_5_H_5_), **48**, which reacted with a range of electrophiles. This concept was further developed by Eyman [42] using the isoelectronic manganese system [(η^6^-C_6_Me_6_)Mn(CO)_3_]^+^, **49**, as exemplified in Figure 11. Moreover, it was found that substitution of a carbonyl ligand by a phosphine, as in [(η^6^-C_6_Me_6_)Mn(CO)_2_PMe_3_]^+^, increases the electron density on manganese and enhances the nucleophilicity of the exocyclic methylene group [43].

To test this approach, [(trindane)Fe(C_5_H_5_)]^+^, **9**, was deprotonated by *t*-BuOK in the presence of excess methyl iodide and yielded a product mixture in which up to twelve benzylic positions had been substituted, as shown in Figure 12. The analogous reaction with allyl bromide also delivered the dodeca-substituted material.

In an attempt to hinder the approach by an electrophile to the *endo* face of the polycyclic ligand, the cyclopentadienyl-iron fragment was replaced by the more voluminous tricarbonylmanganese moiety. However, when [(trindane)Mn(CO)_3_]^+^, **8**, was deprotonated by *t*-BuOK in the presence of excess allyl bromide, a red crystalline product, obtained in 17% yield after chromatographic separation, was identified, surprisingly, as (trindane)Mn(CO)_2_Br, **50** (Figure 13). This material is also readily prepared, in a 93% yield, via the reaction of **8** with trimethylamine N-oxide (to bring about the loss of a carbonyl ligand) and a subsequent reaction with *n*-Bu_4_N^+^Br^−^. Likewise, the reaction of **8** with *t*-BuOK and methyl iodide produced (trindane)Mn(CO)_2_I, **51** (15%). The X-ray crystal structures of **50** and **51**, shown in Figure 15, reveal that, as in (trindane)Cr(CO)_3_, **6,** the tripodal moiety is oriented such that its ligands are staggered with respect to the cyclopentenyl rings, which adopt envelope conformations with *endo*-folded wingtip methylene groups [44].

This contrasts the behaviour of [(C_6_Me_6_)Mn(CO)_3_]^+^, for which the displacement of a carbonyl ligand to form (C_6_Me_6_)Mn(CO)_2_X, where X = Cl, Br or I, requires either photolysis or a reaction with Me_3_NO in the presence of NaX [45]. It is also noteworthy that the treatment of (C_6_Me_6_)Mn(CO)_2_Cl with *t*-BuLi yields (C_6_Me_6_)Mn(CO)_2_H, **52**, apparently via the elimination of isobutene from the presumed *tert*-butyl intermediate (Figure 14). However, this hydride, which is sufficiently stable for its structure to be corroborated via X-ray crystallography [46], is more conveniently prepared via the reaction of (C_6_Me_6_)Mn(CO)_2_I with (*n*-Bu)_4_N^+^ [BH_4_]^−^; furthermore, **52** reacts readily with CCl_4_ or CHCl_3_, but not with CH_2_Cl_2_ to form (C_6_Me_6_)Mn(CO)_2_Cl [47].

In light of these data, one can postulate a mechanism for the formation of (trindane)Mn(CO)_2_I upon the treatment of **51** with *t*-BuOK in the presence of methyl iodide. Since [(C_6_Me_6_)Mn(CO)_3_]^+^ is known to react with methanol to form the rather unstable ester (C_6_Me_6_)Mn(CO)_2_CO_2_Me [48], one can readily envisage the formation of the *tert*-butyl ester, **53**, which readily eliminates isobutene and carbon dioxide via a very favourable six-membered transition state to produce (trindane)Mn(CO)_2_H, **54**, as in Figure 15. Treatment of **8** with *t*-BuOK in CH_2_Cl_2_ furnished (trindane)Mn(CO)_2_Cl, suggesting that **54** is more reactive than (C_6_Me_6_)Mn(CO)_2_H towards alkyl halides.

Since numerous attempts to prepare **54** via the treatment of (trindane)Mn(CO)_2_Br with (*n*-Bu)_4_N^+^ [BH_4_]^−^, or to detect the hydride signal via NMR were unsuccessful, an attempt was made to trap the purported metal hydride either as (trindane)Mn(CO)(PR_3_)H, or as the formyl complex (trindane)Mn(CO)(CHO)(PR_3_). When the cation **8** and *tert*-BuOK were treated with trimethyl phosphite in THF and kept at 40 °C for 20 h, the products isolated after chromatographic separation were yellow crystalline materials **55** and **56**, the ^1^H and ^13^C NMR data of which indicated that the three-fold symmetry of the trindane ligand had been broken, as shown in Figure 16. These products were unambiguously identified via X-ray crystallography as η^5^-indenyl complexes (Figure 16) in which the manganese had migrated from the central arene onto a five-membered ring that had evidently lost three hydrogens. The reaction with triphenylphosphine behaves analogously to give **57** [49].

In seeking a precedent for such behaviour, we note that King reported the reaction of [(η^6^-indane)Mn(CO)_3_]^+^ with *t*-BuOK and a phosphine to yield (η^5^-indenyl)Mn(CO)_2_L, where L = P(OMe)_3_ or PPh_3_ [50]. Furthermore, Crabtree and Parnell found that [(η^6^-indane)Ir(PPh_3)2_]^+^ underwent dehydrogenation and an η^6^-to-η^5^ haptotropic shift to yield [(η^5^-indenyl)Ir(PPh_3)2_(H)]^+^ [51]. Moreover, Ustynyuk and co-workers have shown that (η^5^-indenyl)Cr(CO)_3_Me, **58**, undergoes a “ricochet reaction” in which the methyl is delivered to the five-membered ring and the tricarbonylchromium moiety migrates onto the six-membered ring, probably via an isoindene intermediate, **59**, (Figure 17) [52].

To account for the formation of the η^5^-bonded rearrangement complexes **55** and **56**, we suggest that, in the absence of an alkyl halide, the initially generated (trindane)Mn(CO)_2_H, **54**, undergoes hydrogen migration from an *endo*-benzyl site onto the metal, thus producing the cyclohexadienyl complex, **60**, which in turn loses dihydrogen to give **61**. A second round of *endo*-benzyl hydrogen migration to yield the isoindene framework, **62**, is followed by the final round of hydrogen migration, producing **63**. The loss of dihydrogen and the incorporation of either a carbonyl or phosphite ligand would give the observed products, **55** and **56**, respectively, as depicted in Figure 18 [49].

## 7. The Synthesis of Sumanene

The synthesis of sumanene (Figure 19) was finally achieved by Sakurai, Daiko and Hirao, who prepared a bromo-lithio derivative of norbornadiene, **64**, that was converted into the corresponding tin compound **65**. The copper-mediated trimerisation of **65** gave a 3:1 mixture of *anti* and *syn* benzotris(norbornadiene), **66**, the latter of which underwent metathesis with the Grubbs catalyst to produce hexahydrosumanene, **67**. The process was completed upon oxidation with DDQ to deliver sumanene, **39**, with no need for a flash vacuum pyrolysis step [53]. This work has since been comprehensively reviewed recently [54,55].

## 8. Towards C_60_

### 8.1. From Decacyclene

As we know, the first well-characterised fullerene, C_60_, is now commercially available, but efforts towards a rational stepwise synthetic procedure continue to unfold [36]. As noted above, the chemistry of corannulene and sumanene have been well explored and numerous derivatives have been reported. Nevertheless, other elegant approaches based on three-fold symmetry have been disclosed. In particular, we note work by Scott, a major pioneer in the fullerene field, starting from decacyclene (Figure 20). Having successfully prepared 8-chloro-1(2*H*)-acenaphthylenone, **68**, this was trimerised using TiCl_4_ to produce 3,9,15-trichlorodecacyclene, **69**, which upon FVP treatment at 1100 °C delivered a geodesic dome in a 27% yield with the formula C_36_H_12_, representing 60% of C_60_, and was given the trivial name *Circumtrindene*, **70** [56]. The structure was confirmed via X-ray crystallography [57], and Figure 17 illustrates this molecule as a component of the C_60_ framework.

### 8.2. The Designed Stepwise Synthesis of C_60_

The three-fold cyclisation of ketones, such as that of cyclopentanone to trindane, or indanone to truxene, is mirrored by the route to trichlorodecacyclene, **69**, from 8-chloroacenaphthylenone, **68**. Continuing with this approach, an even more audacious experiment by Scott and de Meijere sought to prepare C_60_ directly via the trimerisation of a C_20_ ketone. As shown in Figure 21, the treatment of the pentacyclic ketone **71** with TiCl_4_ in refluxing *o*-dichlorobenzene led to the *C*_3h_-symmetric molecule, **72**, with the formula C_60_H_27_Cl_3_. They were hoping to effect multiple dehydrohalogenations and dehydrogenations to bring about a “stitching together” of the three arms of the molecule via FVP at 1100 °C, thus forming C_60_ directly. Gratifyingly, this strategy was successful, and the only fullerene formed was C_60_; though in a very low yield, it was sufficient to be unambiguously characterised via mass spectrometry [58].

## 9. Organic Chemistry of Trindane

### 9.1. Oxidation of Trindane

The reported organic chemistry of trindane has focused primarily on its reactivity when treated with different oxidants. Ruthenium trichloride and sodium periodate, in situ, generate a ruthenium(VIII) species, “RuO_4_”, which reacted with trindane to bring about an unanticipated carbohydrate-like product, **72**, the structure of which was elucidated via X-ray crystallography. One can envisage a mechanism whereby one of the double bonds in the aromatic ring is doubly hydroxylated to yield **73**, and the subsequent cleavage of a second double bond forms a diketone that upon further oxidation and hydrolysis rearranges to deliver the final product (Figure 22) [59].

In contrast, the ozonolysis of trindane led to **74**, which again was only unequivocally characterised via X-ray crystallography (Figure 23). Under these conditions, the initial cleavage of two double bonds of the central aromatic ring forms the tetraketone, **75**, that is in equilibrium with its enol tautomer, **76**, and cyclisation leads to the final product. The retention of the C_15_ periphery in **74** resembles the structure of some natural products, such as ginkgolides, and demonstrates that the cleavage of the π-bond *endo* to the cyclopentane present in trindane provides a simple route to complex natural products [60].

More recent work has revealed that the result of the Ru(VIII) oxidation of the next higher homologue of trindane, dodecahydrotriphenylene, **77**, which is readily prepared via the cyclisation of cyclohexanone, differs from that found with trindane. In this case, the central aromatic ring remains intact and the products arise from benzylic oxidation to the mono-, di- and tri-ketones **78**, **79** and **80**, respectively, depending on the length of the reaction time (Figure 24) [61]. Interestingly, the products maintain their directionality such that the ketones are arranged unidirectionally (note that the cyclohexanone rings adopt their normal non-planar geometry).

### 9.2. Synthesis of a Potential Artificial Receptor

In a particularly elegant example that takes advantage not only of the three-fold symmetry of trindane, but also of the ability of an organometallic fragment to block one face of the arene, these concepts were exploited to investigate the construction of artificial receptors. As illustrated in Figure 25, the hexaester **81** was prepared from 1,3,5-tris(bromomethyl)-2,4,6-tris(chloromethyl)benzene via a reaction with the sodium enolate of diethyl malonate, whereupon saponification, decarboxylation and esterification produced a mixture of *cis*,*cis*,*cis*- and *cis*,*cis*,*trans*-trindane-2,5,8-tricarboxylic ester, **82**. However, the treatment of the triester with either Cr(CO)_6_ or Mo(CO)_6_ to form **83** allowed the isolation of the *all-syn* isomer in a good yield. A subsequent reaction with LDA and benzyl bromide proceeded exclusively via *exo* attack to form the desired *C*_3v_-symmetric product. The subsequent removal of the metal carbonyl tripod with iodine to form **84**, and further elaboration led to the urea derivative, **85**, which was investigated for its anion-binding capability [62].

## 10. Applications of Truxene and Truxenone

Truxene, **28**, with its planar *C*_3h_ symmetry, is an important building block for dendrimers, organic frameworks, and star-shaped, cage-like and porous molecules [63,64,65]. An early example of its application was its use in liquid crystals [66], but it has more recently found widespread utility in materials science [67], and in polymer chemistry [68], as a component of the stationary phase in capillary gas phase chromatography [69], as an organic sensitiser, or as a potential hole transport material in perovskite solar cells [70,71].

Truxenone, **86**, was originally prepared directly via the acid-catalysed triple condensation of indane-1,3-dione [72,73], but the more recent five-step synthesis starting from 2-methylacetophenone (Figure 26) proceeds under milder conditions and is more tolerant to sensitive substituents [74].

The ready availability of truxenone has led to the syntheses of a very large number of functionalised derivatives, many of which have been characterised via X-ray crystallography. A common feature that emerges in such structural determinations is the loss of planarity caused by steric crowding between the dicyanomethylene substituents at positions 5, 10 and 15 and the neighbouring peripheral aromatic rings. Typically, as shown in Figure 18, truxene maintains its *C*_3h_ symmetry [75], whereas in the tris(dicyanomethylidene) system, **87**, prepared via Knoevenagel condensation from truxenone (Figure 27), the horizontal plane was broken, and the propeller-type molecule adopted a *C*_3_ symmetry and was chiral [76]. Truxenone has found applications in many areas ranging from organic voltaics [77], and n-type semiconductors [78], to serving as the framework for the cathode in a solid-state lithium-ion battery [79].

## 11. Concluding Remarks

The three-fold *D*_3h_ symmetry of trindane, **1**, or of the trindene trianion, **4**, provides a versatile framework in which chemical structure and reactivity can be controlled. When an organometallic fragment, such as M(CO)_3_, is π-complexed to one face of trindane, it not only enhances the acidity of the *exo*-benzylic hydrogens, but also protects that face from approach by electrophiles. However, nucleophilic attack on the metal carbonyl ligands, with the elimination of specific fragments (such as isobutene or CO_2_), can bring about the formation of a metal–hydride linkage that opens up other reaction possibilities, as exemplified in Figure 16.

When the trindene system bears three π-complexed organometallic units, at least two of them must be sited adjacently on the same face, thereby enhancing the likelihood of electronic interaction between the metals, or even the development of a formal metal–metal bond, as in Figure 8. The addition of three peripheral benzo rings, as in truxene, maintains the three-fold symmetry of the system but also offers different types of coordination sites whereby the organometallic moiety can bind in an η^5^ or η^6^ fashion. Moreover, protonation–deprotonation sequences can bring about haptotropic shifts between these two situations, as exemplified in Figure 6.

The formation of decacyclene apparently regains the *D*_3h_ symmetry of trindenyl, but this is not in fact the case. Non-bonded repulsions between hydrogens of neighbouring naphthyl groups cause the molecule to adopt a chiral (*C*_3_) propeller geometry that is even more evident when organometallic units are attached to opposite faces of the molecule.

It has long been hoped that functionalised versions of these polycyclic frameworks could lead towards a rational stepwise route to fullerenes, or at least to sizeable portions of the C_60_ skeleton. This approach was initially used in early attempts to prepare sumanene, and one spectacular example of success has been the controlled coupling of adjacent peripheral naphthyl rings in decacyclene to generate circumtrindene, C_36_H_12_, which contains 60% of the C_60_ framework.

Finally, one can only marvel at the complexity of the oxidation products of a molecule as simple as trindane, and the multitude of important applications, such as liquid crystals, optoelectronics, solar cell technology or solid-state batteries, in which truxene or truxenone plays such a crucial role.

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
