# Peer review of "Syntheses, Structures and Reactivity of Metal Complexes of Trindane, Trindene, Truxene, Decacyclene and Related Ring Systems: Manifestations of Three-Fold Symmetry"

_molecules, 2023, doi:10.3390/molecules28237796_

Round 1

Reviewer 1 Report

Comments and Suggestions for Authors

Here are some suggestions that would make the manuscript clearer for the benefit of the reader.

Through the manuscript: For all figures with molecular structures shown, if the structure is based on the crystallographic entry from the Cambridge Structural Database (CSD), please use the corresponding CSD ID number in the figure caption (e.g., for Figure 1, structure 6 – NEVVOZ). Otherwise, please specify how the structure was generated.

Page 2, line 54; Page 5 line 141: Structure of [99mTc(trindane)2]+ [PF6]-, 5 is not shown anywhere in the text.

Page 3, line 98: I suggest changing “central methylene” to “central (wingtip) methylene” and using “wingtip methylene” throughout the rest of the text consistently for clarity (e.g., line 105).

Figure 6: for consistency, show the molecular formula on the reaction arrow.

Scheme 4, Scheme 8: please show all the reactant/product structures to be consistent throughout the manuscript.

Page 7, line 208: why is electrochemical behavior relevant here? These two paragraphs seem out of place without further context.

Figures 10 and 11; page 11, line 293: change “bird’s eye view” with “top view”.

Scheme 26: number designation 86 for truxenone is missing.

Page 22, line 604: “substituents at positions 5, 10, and 15” - please illustrate within Figure 18 or Scheme 26/27.

Section 10 - Applications of Truxene and Truxenone, feels short and could benefit from expanding the descriptions on the importance of these applications.

Author Response

Reviewer 1

Through the manuscript: For all figures with molecular structures shown, if the structure is based on the crystallographic entry from the Cambridge Structural Database (CSD), please use the corresponding CSD ID number in the figure caption (e.g., for Figure 1, structure 6 – NEVVOZ). Otherwise, please specify how the structure was generated.

Agreed; an excellent suggestion by the reviewer. We have now included the CSD ID numbers on all X-ray structures. Many of these were our own data and, of course, we had previously deposited them on the Cambridge Structural Database.

Page 2, line 54; Page 5 line 141: Structure of [99mTc(trindane)2]+ [PF6]-, 5 is not shown anywhere in the text.

We did not show this technetium structure when first mentioned for historical purposes (lines 49, 50) as it was only an unverified claim. However, on line 143 and in Figure 5, we do show the structure of the closely analogous ruthenium sandwich complex that we had synthesized and fully characterized; we also pointed out the link between the two molecules.

Page 3, line 98: I suggest changing “central methylene” to “central (wingtip) methylene” and using “wingtip methylene” throughout the rest of the text consistently for clarity (e.g., line 105).

Agreed; changed as suggested on lines 75, 81,99, 102, 106, 110, 114, 131, 407.

Figure 6: for consistency, show the molecular formula on the reaction arrow.

We show the molecular structure, rather than just the formula because this would have been ambiguous and unclear to the reader.

Scheme 4, Scheme 8: please show all the reactant/product structures to be consistent throughout the manuscript.

They are all shown in these Schemes.

Page 7, line 208: why is electrochemical behavior relevant here? These two paragraphs seem out of place without further context.

On lines 209-227 we present the known electrochemical data on trindene complexes in which organometallic fragments are positioned adjacent to each other on a ligand surface. The best way to investigate whether there is electronic interaction between the metals is by using electrochemical probes. We also discuss this phenomenon on lines 625-628 in the Conclusions section.

Figures 10 and 11; page 11, line 293: change “bird’s eye view” with “top view”.

Bird’s eye view is a very commonly used description when viewing such a molecule.

Scheme 26: number designation 86 for truxenone is missing.

The structure and number, 86, of truxenone is shown directly below in Scheme 27.

Page 22, line 604: “substituents at positions 5, 10, and 15” - please illustrate within Figure 18 or Scheme 26/27.

This is now made clear on line 607 where it is noted that the dicyanomethylene substitutents are located at positions 5, 10 and 15, as shown in Scheme 27.

Section 10 - Applications of Truxene and Truxenone, feels short and could benefit from expanding the descriptions on the importance of these applications.

The industrial applications of truxene and truxenone are very numerous and certainly too many for us to discuss in detail. However, we provide links to recent reviews in this area.

Reviewer 2 Report

Comments and Suggestions for Authors

A nice review on an interesting topic. Requires some minor revisions as noted below:

The first line of the abstract would be clearer with some parentheses rather than all commas: “… to trindane (C15H18) or truxene (C27H18), respectively”. Otherwise, it looks like a confusing list.

The title refers only to metal complexes, but there are three sections on their organic chemistry; Sections 7–9. It’s not clear in what way aspects of their organic chemistry was chosen to be included or excluded.

Line 85: “triple-bridged”?

Is it a reasonable assumption that the solid state and solution structures of the trindane complexes are the same, i.e. folded in an endo fashion? If so, why are they folded in this way?

Regarding complexes 36 and 37, it could be noted that the Fe–C distances to the bridging C atoms are longer as these C atoms must each share only one p electron between two metals; then the 18-electron rule is satisfied.

Ensure that compounds are labelled in schemes and figures: Scheme 13 and Figure 15 should label compounds 50 and 50/51, respectively. Similarly for Schemes 16 and 26.

Regarding compound 80, given that this would likely be a planar molecule on the NMR timescale, can one really refer to clockwise and anticlockwise species? That only depends on which side you view it from. Perhaps unidirectional would be better.

Author Response

Reviewer 2

A nice review on an interesting topic. Requires some minor revisions as noted below:

The first line of the abstract would be clearer with some parentheses rather than all commas: “… to trindane (C15H18) or truxene (C27H18), respectively”. Otherwise, it looks like a confusing list.

Agreed; changed as suggested on line 12.

The title refers only to metal complexes, but there are three sections on their organic chemistry; Sections 7–9. It’s not clear in what way aspects of their organic chemistry was chosen to be included or excluded.

It is perhaps surprising that the organic chemistry of these ligands has been so little studied. We report their known reactivity in the hope that it will prompt other researchers to expand this area.

Line 85: “triple-bridged”? Agreed; changed as suggested.

Is it a reasonable assumption that the solid state and solution structures of the trindane complexes are the same, i.e. folded in an endo fashion? If so, why are they folded in this way?

The correlation of NMR coupling constants (in solution) with the dihedral angles found in the X-ray crystallographic data (in the solid state) strongly support the assumption that the structures are the same in both media. The reviewer’s question about the direction of fold is a good one. It is well-established that cyclopentyl rings adopt envelope conformations, and in all these cases the wingtip methylenes point down into the spaces left by the staggered carbonyl ligands. This may prompt a theoretician to further investigate this phenomenon.

Regarding complexes 36 and 37, it could be noted that the Fe–C distances to the bridging C atoms are longer as these C atoms must each share only one p electron between two metals; then the 18-electron rule is satisfied.

Agreed; changed as suggested on lines 294-296.

Ensure that compounds are labelled in schemes and figures: Scheme 13 and Figure 15 should label compounds 50 and 50/51, respectively. Similarly for Schemes 16 and 26.

These are now clearly labelled in the Figure legends.

 Regarding compound 80, given that this would likely be a planar molecule on the NMR timescale, can one really refer to clockwise and anticlockwise species? That only depends on which side you view it from. Perhaps unidirectional would be better. Agreed; changed as suggested, on line 551.

Round 2

Reviewer 1 Report

Comments and Suggestions for Authors

The requested changes/clarifications provided by the authors are satisfactory, I recommend publishing the manuscript in this form.